
# The impact of the diffusion parameter on the passage time of the folding process

**Marcelo Tozo Araujo[1], Jorge Chahine[2], Elso Drigo Filho[2] and Regina Maria Ricotta[3★]**

**1** União das Faculdades dos Grandes Lagos, UNILAGO - São José do Rio Preto-SP
**2** Instituto de Biociências, Letras e Ciências Exatas, IBILCE-UNESP
**3** Faculdade de Tecnologia de São Paulo - Fatec-SP - CEETEPS

★ regina@fatecsp.br

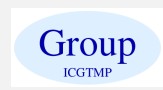
## Abstract

**Recently, a mathematical method to solve the Fokker Plank equation (FPE) enabled the analysis of the protein folding kinetics, through the construction of the temporal evolution of the probability density. A symmetric tri-stable potential function was used to describe the unfolded and folded states of the protein as well as an intermediate state of the protein. In this paper, the main points of the methodology are reviewed, based on the algebraic Supersymmetric Quantum Mechanics (SQM) formalism, and new results on the kinetics of the evolution of the system characterized in terms of the diffusion parameter are presented.**

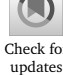
## 1 Introduction

Proteins are structures made up of chains of amino acids. In the unfolded state the protein is in a linear configuration of amino acids and is synthesized in a folded three-dimensional structure to perform specific functions in the organism. To reach this final structure, this linear sequence can pass through intermediate conformations, which are transition states in which the protein may not reach its three-dimensional structure. The importance of studying folding lies in understanding how an unfolded linear structure reaches a three-dimensional and functional, folded structure. The subject has been extensively studied within the last decades under the concept of folding funnel, where the energetic scenario has the shape of a funnel, [1–3].

In a previous work, [4], a consistent mathematical model to physically describe the biological process of protein folding was introduced. The approach considers the process as a diffusion model inspired by the concept of a folding funnel, aiming to analyse its dynamic behavior. The protein folding process is described by the Fokker-Planck equation, FPE, associated to a free energy described by a tri-stable symmetric potential function V(x). In turn, the

FPE can be mapped to a Schrödinger-type equation, SE, [5,6], i. e., both equations share the same spectrum. At this point the methodology of Supersymmetric Quantum Mechanics, SQM, associated with the variational method, [7,8], is used to obtain the approximate spectrum of energy and eigenfunctions of SE and to evaluate the time-dependent probability function $P(x, x_0, t)$, FPE solution, where $x$ is the reaction coordinate, [9], and the coordinate $x_0$ is associated to the protein in the unfolded state.

The free energy, given by the tri-stable potential function $V(x)$, is a symmetric function that has lateral minima with the same depth (symmetric wells) that can be interpreted, respectively, as the folded and unfolded protein states; the central minimum is related to an intermediate protein conformation. The kinects of the diffusion process was characterized by the calculation of the particle population of the right well (folded state). The time required for the evolution of the population of the system from its initial state to the well on the right is used as the characteristic passage time of the system to the folding state of the protein. The results in [4] are consistent with those expected in similar diffusion problems, [10].

In this work a short review of the methodology is presented (Section 2) showing the connection of the FPE with the SE, given in terms of the free energy $V(x)$. The model is illustrated by a specific free energy function, a study case different from the one in [4]. Section 3 contains new results, a mapping of the diffusion dependence and its influence on the symmetric free energy profile performed aiming to analyse the way the increase of the diffusion impacts the passage time to the protein folded state. Section 4 contains the conclusions.

## 2 Methodology: FPE and SE formalism

The probability distribution, FPE's solution, is found by a mapping on an SE, whose solutions are obtained by the variational method associated with the SQM. The free energy to be used is described by tri-stable potentials. Because it is time-dependent, probability distribution describes a characteristic time for the dynamics of protein folding through an estimate of the passage time as a function of the reaction coordinate. The behavior of the population towards the third well is verified, which characterizes the folded state, as a function of time, as an exponential decay characteristic of diffusive processes with a directional force.

The FPE, which describes the time evolution of the probability distribution P(x,t) in diffusion systems is given by

$$\frac{\partial}{\partial t} P(x,t) = -\frac{\partial}{\partial x}[f(x).P(x,t)] + Q\frac{\partial^2}{\partial x^2}P(x,t), \tag{1}$$

where $x$ is the characteristic variable of the system, the reaction coordinate (number of native contacts); t is the time variable; $Q$ is the diffusion coefficient and $f(x)$ represents an external force (driving force) acting on the medium, it is associated with the free energy of the medium, the tri-stable potential $V(x)$,

$$f(x) = -\frac{d}{dx}V(x). \tag{2}$$

Writing the probability P(x,t) as a product of a function of x and a function of t,

$$P(x,t) = \Psi(x)e^{-\lambda t}, \tag{3}$$

it can be shown that the FPE solutions are solutions of a time-independent Schrödinger-type equation, SE, given by

$$\frac{d^2}{dx^2}\Psi(x) - \frac{1}{2Q}\left(\frac{f(x)^2}{2Q} + \frac{df(x)}{dx}\right)\Psi(x) = \frac{\lambda}{Q}\Psi(x), \tag{4}$$

where $\lambda$ is proportional to the energy. Expanding $\Psi(x)$ on an orthonormal basis, we obtain the probability distribution given by

$$P(x,t|x_0,t_0) = \frac{\Psi_0(x)}{\Psi_0(x_0)} \sum_{n=0}^{\infty} \Psi_n(x) \cdot \Psi_n(x_0) \cdot e^{-\lambda_n(t-t_0)}, \tag{5}$$

where $\Psi_0(x)$ is the ground state wave function and $x_0$ is the starting position. The SE used by SQM is expressed, in reduced units, in general as

$$-\frac{d^2}{dx^2}\Psi(x) + \underbrace{\left(W_1(x)^2 - \frac{dW_1(x)}{dx} + E_0^{(1)}\right)}_{V_{SE}(x)}\Psi(x) = E\,\Psi(x), \tag{6}$$

where $V_{SE}(x)$ is the Schrödinger potential function defined in terms of the superpotential function $W_1(x)$, [7]. Comparing the equations (4) and (6) and considering the relationship of $f(x)$ with $V(x)$ given by Eq. (2), we obtain

$$W_1(x) = \frac{1}{2Q}\frac{dV(x)}{dx}, \tag{7}$$

that is, the FPE potential $V(x)$ is related to the superpotential $W_1(x)$ of SQM. Also the energy $E$ is related to the parameter $\lambda$ as

$$E = \frac{\lambda}{Q}. \tag{8}$$

Thus, the SQM methodology associated with the variational method can be used to determine the spectrum, [8]. At this point it is important to remark the relationship between potential function of the SE (6) with the diffusion parameter $Q$, through the superpotential $W_1$ in (7),

$$V_{SE}(x) = W_1(x)^2 - \frac{dW_1(x)}{dx} + E_0^{(1)}. \tag{9}$$

In other words, when using the SQM methodology the spectrum is explicitly dependent on the value of the diffusion constant Q.

## 2.1 The tri-stable potential and the spectrum

The tri-stable potentials used are of the type

$$V(x) = ax^6 - 8.93851x^4 + 5.42373x^2, \tag{10}$$

illustrated in Figure 1 for various values of the constant $a$. The lateral minima ($V_{min}$) have the same depth (symmetrical wells) and are interpreted, respectively, as the unfolded (left well) and folded (right well) states of the protein, and the central minimum is related to a set of intermediate protein conformations, with $\Delta V = V(0) - V_{min}$. The choice of parameters was made in order to have several symmetrical potentials with V(0)=0 and different lateral depths of the wells, as in [10]. Thus, only the variation of the parameter $a$ in each tri-stable potential is enough to deal with the depth of the lateral minima.

## 2.2 Study case for fixed diffusion parameter Q

To illustrate the model, we choose the potential $V(x)$ with $a = 3.90456$ and fixed diffusion constant, $Q = 0.5$, as in Figure 2. The value of Q is arbitrary but it has to be fixed in order to apply the SQM methodology to obtain the spectrum, as it can be seen from the superpotential $W_1$ in equation (7). In Section 3 we vary the diffusion constant and evaluate its impact on the

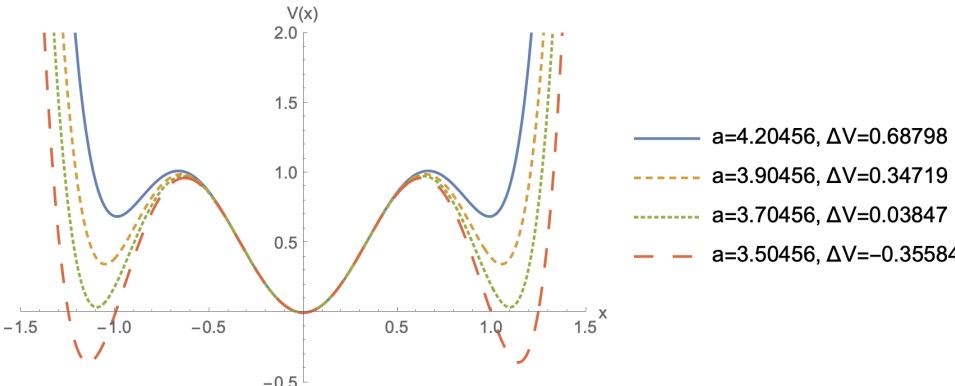

Figure 1: Representation of tri-stable potentials, Eq.(10) for different values of the constant $a$ with the respective values of $\Delta V = V(0) - V_{min}$, [4].

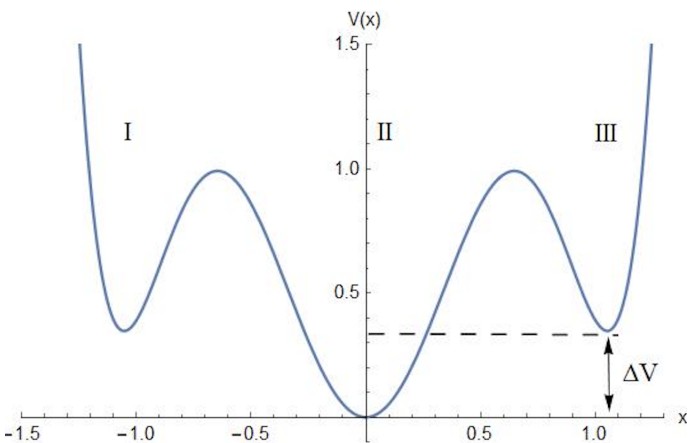

Figure 2: Representation of tri-stable potential function, $V(x) = 3.90456x^6 - 8.93851x^4 + 5.42373x^2$ with the respective value of $\Delta V = 0.34719$, [4].

folding kinetics. It should be mentioned that in other works, [10], the quantities are given in units $\frac{1}{Q}$.

Once defined the free energy given by the potential $V(x)$ we return to the construction of the equivalent SE spectrum, solution of Eq. (6). Then using the SQM methodology, [7]- [8], the approximate spectrum of energies and eigenfunctions is shown by Table 1 and Table 2. It should be stressed that as we are dealing with an approximative method, the number of terms in the probability expansion, Eq. (5), was fixed to six terms in the series, ($n = 0, ..., 5$), since that the contribution of the next exponential term is several orders of magnitude smaller than the previous term and thus can be neglected.

From the SE spectrum, the probability density, given by Eq. (5), can be calculated for different starting points $x_0$. The diffusion process is then characterized by calculating the population defined by

$$\mathcal{N}(t) = \int_{x_i}^{x_f} P(x,t)dx\,, \tag{11}$$

where the limits of integration $x_i$ and $x_f$ refer to the investigation region of the particle population, regions I, II and III, as denoted in Figure 2.

Table 1: Values of the energy spectrum of the SE when the potential function (free energy) is $V(x) = ax^6 + bx^4 + cx^2$ with the values $a = 3.90456$, $b = -8.93851$, $c = 5.42373$.

| n | 0 | 1 | 2 | 3 | 4 | 5 |
|---|---|---|---|---|---|---|
| $\lambda_n$ | 0 | 1.0361 | 1.9797 | 9.2517 | 17.3589 | 27.4762 |

Table 2: Wave functions spectrum of the SE, when the potential function (free energy) is $V(x) = ax^6 + bx^4 + cx^2$ with the values $a = 3.90456$, $b = -8.93851$, $c = 5.42373$.

$$\Psi_0^{(1)}(x) = 1.021 e^{(-5.42373x^2 + 8.93851x^4 - 3.90456x^6)}$$

$$\Psi_1^{(1)}(x) = e^{(2.73824x^2 - 3.05175x^4 - 0.777117x^6)} x(1.99656 - 8.75317x^2 + 10.4419x^4)$$

$$\Psi_2^{(1)}(x) = e^{(-3.68104x^2 + 1.08564x^4 - 2.13171x^6)}$$
$$(-0.904686 + 5.15415x^2 - 9.43439x^4 + 33.953x^6 - 199.101x^8 + 303.278x^{10})$$

$$\Psi_3^{(1)}(x) = e^{(-3.50991x^2 - 1.40729x^4 - 0.64738x^6)}(-5.23999x + 1.07696x^3 + 20.5988x^5 -$$
$$74.5237x^7 + 66.0048x^9 - 36.7167x^{11} + 73.6961x^{13} + 69.4599x^{15})$$

$$\Psi_4^{(1)}(x) = e^{(-4.601x^2 - 1.49668x^4 - 0.559156x^6)}(0.928068 - 17.7878x^2 + 1.05918x^4 + 5.10868x^6$$
$$-84.9985x^8 - 0.86348x^{10} - 24.6105x^{12} + 118.086x^{14} + 180.684x^{16} + 126.316x^{18}$$
$$+44.1023x^{20})$$

$$\Psi_5^{(1)}(x) = e^{(-5.52086x^2 - 1.54808x^4 - 0.53713x^6)}(4.7712x + 0.962939x^3 - 63.0876x^5 - 155.71x^7 -$$
$$181.732x^9 + 13.8384x^{11} + 423.621x^{13} + 738.097x^{15} + 712.503x^{17} +$$
$$438.794x^{19} + 175.5x^{21} + 42.7624x^{23} + 5.15394x^{25})$$

Figure 3 illustrates the results of the numerical calculation of the population of the left well, region I, $\mathcal{N}_I(t)$ as a function of time t and the population of the right well, region III, $\mathcal{N}_{III}(t)$ for the initial value $x_0 = x_{min} = -1.05282$. The initial population $\mathcal{N}_I(t)$ decreases in time until it reaches equilibrium while the population $\mathcal{N}_{III}(t)$ increases in time until reaching the same equilibrium, revealing the diffusion behavior of the process since the wells are symmetrical.

Figure 4 illustrates the best numerical fit of population versus time t of Region I, $\mathcal{N}_I(t)$, as a function of time t and the population of the well on the right, Region III, $\mathcal{N}_{III}(t)$, for the initial value $x_0 = x_{min} = -1.05282$, numeric data from Figure 3. The fit is given by a decreasing exponential (dotted line) and an increasing exponential (dashed line) with characteristic times: $\tau' = 0.787511$ and $\tau = 1.46159$, respectively. The characteristic time is interpreted as the transition time from region I to region III.

## 2.3 Results for the passage time for different potentials

Figure 5 illustrates the passage time $\tau$ versus the initial position $x_0$ for different potentials $V(x)$ illustrated in Figure 1, revealing a decrease in the value of $\tau$ as a function of the initial position $x_0$, in addition to a decrease in the value of $\tau$ with an increase of $\Delta V$, of the depth of the symmetrical wells.

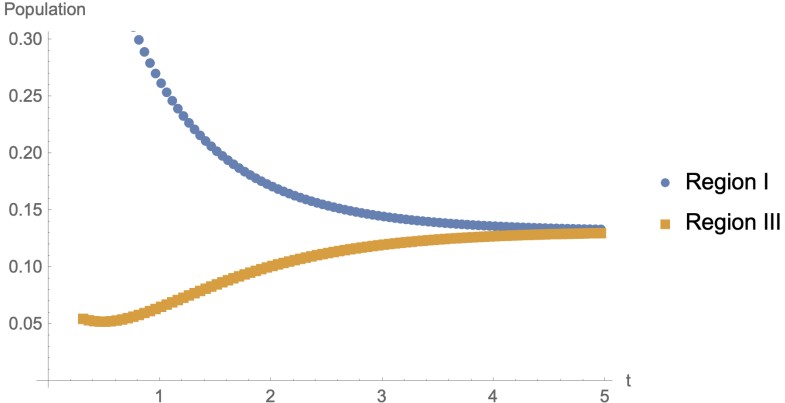

Figure 3: Graph of the population of region I (circle) as a function of time, $\mathcal{N}_I(t)$, and of the population of region III (square) as a function of time, $\mathcal{N}_{III}(t)$, calculated numerically, [4].

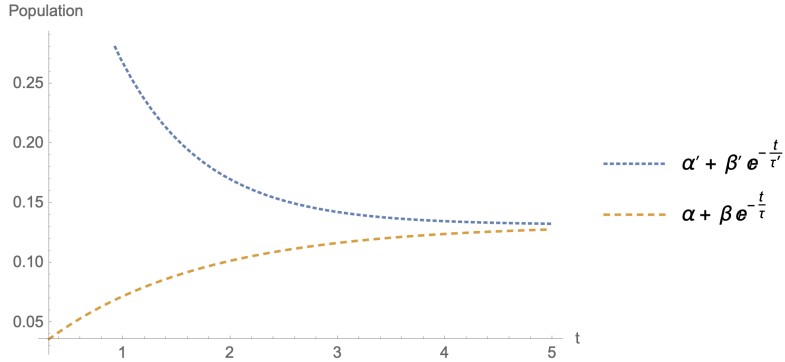

Figure 4: Best numerical fit of population versus region t time of region I, $\mathcal{N}_I(t)$, (dotted line) and the population versus time t of the region III, $\mathcal{N}_{III}(t)$, (dashed line), [4].

## 3 Diffusion

Using the methodology developed in [4] for the protein folding process, the characteristic passage time $\tau$ was evaluated for different values of the diffusion parameter Q for the free energy of Figure 2, $V(x) = 3.90456x^6 - 8.93851x^4 + 5.42373x^2$. For each fixed value of Q in the interval $0.4 < Q < 20$, the passage time $\tau$ for the evolution of the population to the right well was calculated, starting from the initial position $x_0 = x_{min} = -1.05282$, as shown in Figure 6 (dotted line). Figure 6 also shows the best fitting for the results (solid line), given as as function of $1/Q$.

Figure 6 shows that the passage time $\tau$ decreases as the diffusion increases, as expected. The general behavior of the curve of $\tau$ *versus* Q is a function proportional to $1/Q$ which is compatible with that obtained by another method, that uses the stationary state approximation, [10].

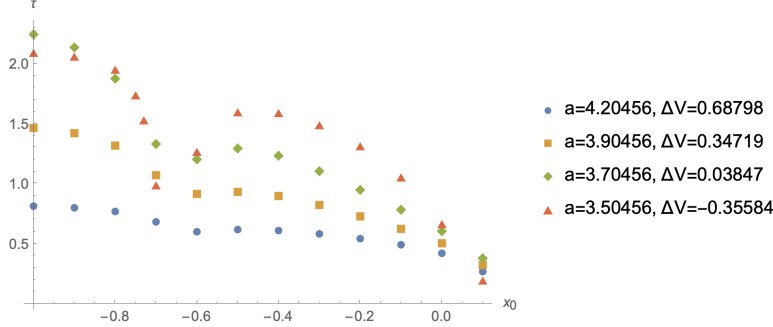

Figure 5: Passage time $\tau$ versus the initial position $x_0$ for the different potentials $V(x)$, as in Figure 1, with $Q = 0.5$, [4].

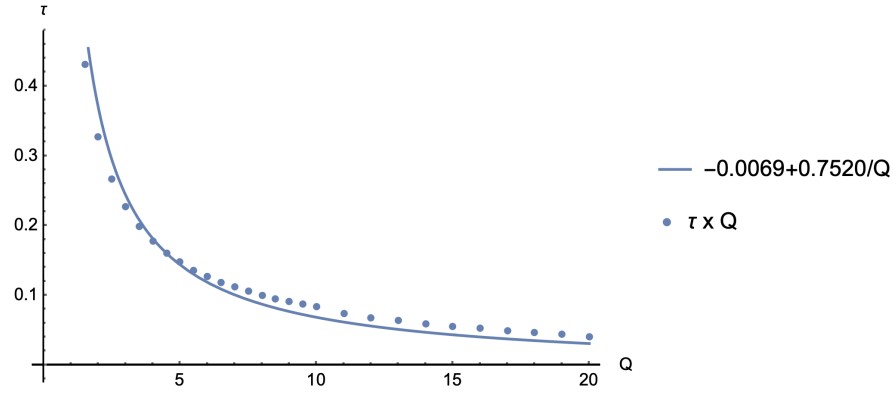

Figure 6: Passage time $\tau$ versus the diffusion constant Q for $V(x) = 3.90456x^6 - 8.93851x^4 + 5.42373x^2$ (Figure 2) for analytical results (dotted line) and the best fitting (solid line).

## 4 Conclusion

The main point of this paper was to determine explicitly the passage time ($\tau$) dependence on the diffusion parameter $Q$. The general $\tau$ *versus* $Q$ curve obtained (Figure 5) is a function proportional to $1/Q$ which is compatible with that obtained by another method, [8].

The results obtained reinforce the application of the SQM mathematical method proposed for protein folding problems, mainly in the determination of P(x,t) by solving the FPE through its relation with the SE. The passage time of the unfolding-folding process is an important ingredient for the reaction kinetics; the results are consistent with those obtained in [8]. In this reference only the SE ground state is used which makes the probability density depend only on $x$ and not explicitly on $t$, i.e., their method only use the stationary state. Our approach allows the use of more terms in the expansion of Eq. (5) which makes the dependence of $t$ on $P(x,t)$ appear explicitly.

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
