# Peer review of "The impact of the diffusion parameter on the passage time of the folding process"

_SciPost Physics Proceedings, doi:SciPost Phys. Proc. 14, 015 (2023)_

## Round 1 · Referee Report · Anonymous (Referee 1) · 2023-3-22

Report
In this work, the authors present an application of the Fokker-Planck equation to model the diffusion process during protein folding. The authors use a potential V that mimics a protein free energy profile with 3 states (unfolded, intermediate, and folded). Nonetheless, there is a lack of discussion in associating this work with protein folding. Comparison with experimental results should be performed. The language and writing should be revised. Breaking sentences would help the reader. There are also major scientific points that should be discussed:
There is a lack of references, especially discussing seminal work from Dmitrii E. Makarov, Robert B. Best, Gerhard Hummer, and others.
There is also a lack of references about the protein folding problem, particularly in the introduction.
Section 2.1 - How do the authors justify the choice of parameters in equation 10?
Section 2.2:
The same, Is there any argumentation for using those values of a and Q?
Why choose a potential V(x) where the intermediate is more stable than the native state? I recommend doing the same calculations for a scenario where the protein's native state is more stable than the intermediate.
What happens when d(V(x)/dx = 0? In this case, the force is zero. How does the protein decide to go right or left in the potential? Do authors consider the case of half o the time, when reaching this inflection point, the protein returns to the previous state?
Why the potential in the native state goes up? What exists after the native state?
The authors mentioned that the reaction coordinate x is the number of native contacts (page 2). How is possible to have negative values of x? The minimum value must be 0. A negative number of contacts is inconsistent.
How these calculations are different from a partible diffusing under the influence of the potential V(x)?
Section 3:
There is a lack of physical units in all calculations and figures. What are the units of V, f, Q, t?
What are the criteria to consider the protein folded when calculating the passage time?
Section 4:
There is a lack of discussion, and the conclusion seems like a textbook exercise. I recommend exploring different parameters and connecting the results with experiments or even simulations.
There is a lack of references, especially discussing seminal work from Dmitrii E. Makarov, Robert B. Best, Gerhard Hummer, and others.
There is also a lack of references about the protein folding problem, particularly in the introduction.
Section 2.1 - How do the authors justify the choice of parameters in equation 10?
Section 2.2:
The same, Is there any argumentation for using those values of a and Q?
Why choose a potential V(x) where the intermediate is more stable than the native state? I recommend doing the same calculations for a scenario where the protein's native state is more stable than the intermediate.
What happens when d(V(x)/dx = 0? In this case, the force is zero. How does the protein decide to go right or left in the potential? Do authors consider the case of half o the time, when reaching this inflection point, the protein returns to the previous state?
Why the potential in the native state goes up? What exists after the native state?
The authors mentioned that the reaction coordinate x is the number of native contacts (page 2). How is possible to have negative values of x? The minimum value must be 0. A negative number of contacts is inconsistent.
How these calculations are different from a partible diffusing under the influence of the potential V(x)?
Section 3:
There is a lack of physical units in all calculations and figures. What are the units of V, f, Q, t?
What are the criteria to consider the protein folded when calculating the passage time?
Section 4:
There is a lack of discussion, and the conclusion seems like a textbook exercise. I recommend exploring different parameters and connecting the results with experiments or even simulations.

Anonymous on 2023-03-22 [id 3502]
In this work, the authors present an application of the Fokker-Planck equation to model the diffusion process during protein folding. The authors use a potential V that mimics a protein free energy profile with 3 states (unfolded, intermediate, and folded). Nonetheless, there is a lack of discussion in associating this work with protein folding. Comparison with experimental results should be performed. The language and writing should be revised. Breaking sentences would help the reader. There are also major scientific points that should be discussed:
There is a lack of references, especially discussing seminal work from Dmitrii E. Makarov, Robert B. Best, Gerhard Hummer, and others.
There is also a lack of references about the protein folding problem, particularly in the introduction.
Section 2.1 - How do the authors justify the choice of parameters in equation 10?
Section 2.2:
The same, Is there any argumentation for using those values of a and Q?
Why choose a potential V(x) where the intermediate is more stable than the native state? I recommend doing the same calculations for a scenario where the protein's native state is more stable than the intermediate.
What happens when d(V(x)/dx = 0? In this case, the force is zero. How does the protein decide to go right or left in the potential? Do authors consider the case of half o the time, when reaching this inflection point, the protein returns to the previous state?
Why the potential in the native state goes up? What exists after the native state?
The authors mentioned that the reaction coordinate x is the number of native contacts (page 2). How is possible to have negative values of x? The minimum value must be 0. A negative number of contacts is inconsistent.
How these calculations are different from a partible diffusing under the influence of the potential V(x)?
Section 3:
There is a lack of physical units in all calculations and figures. What are the units of V, f, Q, t?
What are the criteria to consider the protein folded when calculating the passage time?
Section 4:
There is a lack of discussion, and the conclusion seems like a textbook exercise. I recommend exploring different parameters and connecting the results with experiments or even simulations.
Anonymous on 2023-03-10 [id 3469]
This work was presented in the context of the 34th International Colloquium on Group Theoretical Methods in Physics Strasbourg, 18-22 July 2022, according to the authors.
My comments are as follows: * There is neither mention or relevance to group theory * The connection to protein folding is vague, at least much less than the work quoted in reference: Reference 2 by Wolynes. * The paper is very similar to Réference [1] by the same authors. * The scientific level is weak, a simple exercise based on the diffusion equation (1), no new finding.
The paper should not be accepted in SciPost.

---

## Round 1 · Referee Report · Anonymous (Referee 2) · 2023-4-14

Strengths
The mathematical formalism and biophysical approach
Weaknesses
lack of discussion, and references
Report
The Authors have applied the Fokker-Plank equation to the diffusion model during the protein folding process using a random potential V that represents 3 protein energy state profiles: unfolded, intermediate/transition, and folded, according to my understanding. The mathematical formalism and biophysical approach in this work is very sophisticated and highlighted the need of research in this area.
However, a few points should be addressed:
Introduction: There is not many references for the topic, especially around the protein folding issue and its deregulation and the consequences in the medical field. The author could explore this area in more detail and show data for some diseases and how this work can help to address those biological questions.
Methodology and Formalism:
In general there is a lack of physics parameters units, such as for V, Q, t and f.
On the equation 10, where do the constants -8.93851 and +5.42373 come from? Are they arbitrary or coming from a simulation? Same for the Figure 2 where “a” was fixed as 3.90456. It should be shown how those numbers were fixed, or mentioned if it was arbitrary.
Is “a” and “” (alpha) the same variable? Could that be a typo?
What is the definition of N(t) shown in the equation 11, is it the diffusion process and its distribution? A better variable definition is required.
t and t’ were mentioned on the section 2.2 and defined only in the section 2.3 as passing time and then the same variable is defined as passage time. Please define the variable and its units before it appears in the equations, and keep it consistent throughout the paper.
All the figures (graphics) need to have a bigger x and y variables allocated to its axis for better visualisation. Such as “t x Q” or “t x Xo” or “Population x t”.
In this work there is a lack of discussion, for example the authors could explore how this model could address the Levinthal’s protein folding paradox for (B Bagchi, 1992). Also, there is room to discuss how this Fokker-Plank equation to diffusion model can impact biomedical or biophysics research or techniques, as a novelty proposed in the introduction.
Overall, this is good paper and addresses a part of biophysics research where research is lacking and could bring significant breakthrough in science and medical research. I would recommend it for publication after the authors checked the points raised above.
However, a few points should be addressed:
Introduction: There is not many references for the topic, especially around the protein folding issue and its deregulation and the consequences in the medical field. The author could explore this area in more detail and show data for some diseases and how this work can help to address those biological questions.
Methodology and Formalism:
In general there is a lack of physics parameters units, such as for V, Q, t and f.
On the equation 10, where do the constants -8.93851 and +5.42373 come from? Are they arbitrary or coming from a simulation? Same for the Figure 2 where “a” was fixed as 3.90456. It should be shown how those numbers were fixed, or mentioned if it was arbitrary.
Is “a” and “” (alpha) the same variable? Could that be a typo?
What is the definition of N(t) shown in the equation 11, is it the diffusion process and its distribution? A better variable definition is required.
t and t’ were mentioned on the section 2.2 and defined only in the section 2.3 as passing time and then the same variable is defined as passage time. Please define the variable and its units before it appears in the equations, and keep it consistent throughout the paper.
All the figures (graphics) need to have a bigger x and y variables allocated to its axis for better visualisation. Such as “t x Q” or “t x Xo” or “Population x t”.
In this work there is a lack of discussion, for example the authors could explore how this model could address the Levinthal’s protein folding paradox for (B Bagchi, 1992). Also, there is room to discuss how this Fokker-Plank equation to diffusion model can impact biomedical or biophysics research or techniques, as a novelty proposed in the introduction.
Overall, this is good paper and addresses a part of biophysics research where research is lacking and could bring significant breakthrough in science and medical research. I would recommend it for publication after the authors checked the points raised above.
Requested changes
All the figures (graphics) need to have a bigger x and y variables allocated to its axis, for a better visualisation. Such as “t x Q” or “t x Xo” or “Population x t”.

---

## Round 2 · Author Response

The article presents a summary of a theoretical methodology (Reference [4]) and new results (Section 3). The method concerns the use of the variational method associated with Supersymmetric Quantum Mechanics, which is essentially an algebraic method of quantum mechanics, applied here to provide a theoretical model to analyze the protein folding process.

---

## Round 2 · List of Changes

- The article presents a summary of a theoretical methodology already published (Reference [4]) and new results (Section 3). It is a mathematical model built to understand the protein folding process by solving the Fokker Plank equation. Through a known mapping between this equation and a Schrödinger-type equation, both equations have the same spectrum. Thus, the approximate spectrum is solved through the algebraic method of supersymmetric quantum mechanics (SQM) which allows obtaining the time dependent probability distribution, solution of the the Fokker Planck equation for a given free energy V(x). This allows the subsequent analysis of the (theoretical) kinetics of the protein folding. It is therefore an initial model for the protein folding process. Thus, in future work (in progress) the free energy with specific parameters and units obtained from the computational simulation will be treated and the results compared.
- The Abstract and Introduction were rewritten in order to make clear the points raised by the reviewers. New references were added, References [2] and [3]. We were careful to ilustrate the model in Section 2 with results different from those published in reference [4] (different free energy V(x)).
- Questions concerning the choice of parameters in Equation (10) are explained below it, in Section 2.1 with the addition of the following: "The choice of parameters was made in order to have several symmetrical potentials with V(0)=0 and different lateral depths of the wells, as in Reference [10]. Thus, only the variation of the parameter "a" in each tri-stable potential is enough to deal with the depth of the lateral minima."
- Question concerning the choice of parameter Q is explained in Section 2.2 with the addition of the following: "To illustrate the model, we choose the potential V (x) with a = 3.90456 and fixed diffusion (arbitrary) constant, Q = 0.5, as in Figure 2. The value of Q is arbitrary but it has to be fixed in order to apply the SQM methodology to obtain the spectrum, as it can be seen from the superpotential W1 in equation (7). In Section 3 we vary the diffusion constant and evaluate its impact on the folding kinetics. It should be mentioned that in other works, [10], the quantities are given in units 1/Q."
- The new results are presented in Section 3. The first paragraph was rewritten, to avoid confusion between the time t and the passage time (τ). "Using the methodology developed in [4] for the protein folding process, the characteristic passage time τ was evaluated for different values of the diffusion parameter Q for the free energy of Figure 2, V (x) = 3.90456x6 − 8.93851x4 + 5.42373x2. For each fixed value of Q in the interval 0.4 < Q < 20, the passage time τ for the evolution of the population to the right well was calculated, starting from the initial position x0 = xmin = −1.05282, as shown in Figure 6 (dotted line). Figure 6 also shows the best fitting for the results (solid line), given as function of 1/Q."

---

## Editorial Decision

published